# Insight on Corrosion Prevention of C1018 in 1.0 M Hydrochloric Acid Using Liquid Smoke of Rice Husk Ash: Electrochemical, Surface Analysis, and Deep Learning Studies



Agus Paul Setiawan Kaban [1], Johny Wahyuadi Soedarsono [1], Wahyu Mayangsari [2], Mochammad Syaiful Anwar [2], Ahmad Maksum [3], Aga Ridhova [2] and Rini Riastuti [1,*]

[1]   Prof. Johny Wahyuadi Laboratory, Department of Metallurgical and Materials Engineering, Universitas Indonesia, Depok 16424, Indonesia
[2]   Research Center for Metallurgy—National Research and Innovation Agency, Kawasan Sains dan Teknologi BJ Habibie, Central Jakarta 10340, Indonesia
[3]   Department of Mechanical Engineering, Politeknik Negeri Jakarta, Jl. Professor Doktor G.A. Siwabessy, Depok 16425, Indonesia
*   Correspondence: riastuti@metal.ui.ac.id

**Abstract:** This work reports the anti-corrosion behavior of liquid smoke from rice husk ash to unveil the contribution of its active compounds in 1 M HCl solution. In this study, the developed methodology to test, analyze, and model the novel type of green corrosion inhibitor for C1018 was characterized using Electrochemical impedance spectroscopy (EIS), Potentiodynamic polarization, and deep learning methods. The inhibitor structure was characterized by Fourier transform infrared analysis (FTIR) and Ultraviolet–visible spectroscopy (UV-Vis). The surface characterization of mild steel immersed in blank and 80 ppm solution inhibitor was performed using Atomic force microscopy (AFM) analysis. The corrosion test results show that the inhibitor is considered a mixed-type inhibitor to achieve the optimum inhibition of 80 ppm at 323 K, reaching up to 99% inhibition efficiency. The AFM results show a smoother surface given a lower skewness parameter at $-0.5190$ nm on the treated mild steel. The artificial neural network demonstrates the lower overfitting on the inhibited steel, a higher accuracy prediction of 81.08%, and a lower loss rate of 0.6001 to model the relationship between the EIS and Potentiodynamic polarization and the evolution of the passive layer on the treated mild steel. The experiment agrees well with the prediction result to model the adsorbed inhibitor. The work can be used as a guideline to pave the way for subsequent applicability in developing green corrosion inhibitors based on experimental and artificial intelligence approaches.

**Keywords:** green corrosion inhibitor; liquid smoke inhibitor; deep learning corrosion inhibition

## 1. Introduction

A pipeline's inherent essential role is in delivering oil and gas from the well to the platform, where its primary construction materials are composed of mild steel. The ultimate selection of this material is due to its availability, low cost, and high strength [1]. Commonly, the pipelines are composed of mild steel and ferrous metal with superior mechanical properties. Moreover, the materials are bendable and quickly form a specific shape without losing their structural integrity [2]. Nonetheless, the material remains susceptible to corrosion, leading to critical issues such as a dramatic physical appearance, mechanical properties, and material resistance. A recent publication by the National Association of Corrosion Engineers (NACE) reports that the business losses caused by corrosion are nearly 5% of the country's gross national product [3]. The consequence of the corrosion degrades the pipelines when metal interacts with acidic environments, particularly in the process of chemical (acidic) cleaning [4]. Hydrochloric acid (HCl) is often used to clean the metal of undesired substances (corrosion products) such as stains and inorganic contaminants [5].

The cleaning process helps to restore the flow assurance in the pipeline of oil and gas, which are threatened due to the deposition of corrosion products. The minimal cost and high solubility of HCl in $FeCl_3$ have become made it the primary selection of acid to remove corrosion products. Nevertheless, under such conditions, the pipelines and the metallic equipment exposed to the acid experience severe corrosion.

The strategy to lower the corrosion effect in the pipeline while preserving the environment is by dissolving the green corrosion inhibitor [6–8]. Several publications recently reported using plant extracts as green corrosion inhibitors for mild steel under a lower pH solution. The impact of Rollinia occidentalis extract [9] is practical for reducing the electrochemical activity under an acidic solution. They reported that the active molecules of the extract adsorbed on the mild steel obey the Langmuir adsorption isotherm as the inhibitor affects the cathodic and anodic regions. The work of [10] reveals that in the presence of Curcuma xanthorrhiza extract, the API 5L X42 steel corrosion rate is remarkably reduced through weak electrostatic interaction between the adsorbate and substrate. Similar to the previous study, the inhibition adsorption mechanism follows the Langmuir adsorption isotherm. The publication of [11] studied how the corrosion rate of mild steel decreased in the solution of hydrochloric acid containing the extract of Areca flower. It shows that the corrosion rate is dramatically depressed at 99% when a higher concentrated inhibitor is added to the solution.

Despite the wide-ranging utilization of natural plants as corrosion inhibitors, it is acknowledged that the challenge in developing green inhibitors relates to the low inhibition efficiency at elevated temperatures. It is generally accepted that high temperature quickens the corrosion rate to degrade the metal associated with the thermodynamic process. It transforms metal into its corrosion product, such as $Fe_2CO_3$, because of the effect of electrochemical and thermodynamical reaction inverse metal extraction [12]. Another challenging factor is related to the economic problem of the feedstock of inhibitors to meet a continuous supply for inhibitor production using recycled natural waste.

A few studies report the potential of rice husk ash (RHA) waste to meet wide-ranging applications, such as producing activated carbon [13] and generating electricity [14]. The production of rice husk is given from the milling process, which decreases by 80% the weight of nearly 200 kg/tonnes of rice at high temperatures. It is also possible to note that the present work aims to use and harness the potential of RHA as a green corrosion inhibitor to protect C1018 metal in an HCl medium. The RHA has unique properties comprising several essential compounds, such as 2-methyl-phenol and octadecamethyl-cyclononasiloxane, that contribute to the adsorption process on the surface of metals [15]. In addition, liquid smoke from organic compounds has been recognized for its antioxidant properties [16] due to the presence of phenolics, including carboxyl and alcohol [17], and improves anti-corrosion efficiency. In this work, integrating the distinct properties of liquid smoke into RHA can prepare environmentally friendly corrosion inhibitors and maximize the inhibition. Therefore, it motivates this work to use a greener product with inexpensive raw materials and a feasible process to control metal corrosion.

The primary consideration of the selected liquid smoke of RHA as a corrosion inhibitor is related to the higher number of heteroatoms, the number of pi-electrons, and the presence of electron-rich atoms such as oxygen to form a passive film through a chemical bonding between the 3D vacant orbitals of Fe and inhibitor molecules. Hence, the above-predicted anti-corrosion mechanism is expected to increase the capability of the inhibitor to cover a larger surface area of protection with higher solubility in water and stability at elevated operational temperatures.

On the other hand, the existing engineering science entails an explainable principle, that may or may not cover the application of artificial intelligence. Specifically, developing a deep learning model in corrosion mitigation has not been fully explored. Deep learning is the subset of machine learning that has been used successfully across various applications, including corrosion detection [18] and the risk assessment of pipelines [19]. Inclusively in the field of corrosion science, the application of artificial intelligence has been found on many fronts. Artificial neural networks (ANNs) have gained popularity in classifying gas/liquid/and pulp fiber flow regimes, as published in [20]. In this work, ANN is utilized to seek the relationship between the electrochemical measurement data and the evolution of passive film. Such restrictions to incorporating data science in assessing the effect of increasing the temperature and concentration of corrosion inhibitors allow scientists and engineers to justify experimental results accurately and quickly and free from human subjectivity [21].

Hence, the objective of the paper is to: (i) discuss the inhibition performance of the liquid smoke of RHA on C1018 under an acidic environment; (ii) evaluate the adsorption process related to surface modification, the functional group's identification, and electronic transition; (iii) model the relationship between the electrochemical result and the growth of film thickness using deep learning to quicken the analysis related to the compatibility of the inhibitor under specific conditions. The limitations of this paper include the contribution of RHA major compounds on mild steel C1018 despite multiple derivative molecules that may appear in the FTIR and UV-Vis and have been identified [15]. Moreover, the work excludes the discussion on adsorption isotherm inhibitors since it comprises more than two major compounds. Moreover, this paper restricts the use of ANNs to the classification task to distinguish thin from thick adsorbed inhibitor layers to study the growth of passive film inhibitors based on electrochemical tests.

## 2. Related Work

The issue of metal corrosion in offshore pipelines is critical and causes severe damage to the integrity of the material by reaction with its marine environment. It is common practice to monitor the corrosion rate through onsite corrosion coupons using C1018, particularly in the oil and gas industry [22]. The study of [23] reports that microbiologically influenced corrosion (MIC) in the subsea environment becomes a significant challenge due to the presence of biofilm on the C1018 carbon steel, causing MIC pitting induced by the sulfate-reducing bacteria (SRB). In their work, the corrosion mitigation scenario to protect the metal is carried out by controlling the $H_2S$ concentration using the SRB strain of Desulfovibrio vulgaris.

C1018 mild steel is a low-grade carbon steel material that reacts and corrodes under an acidic environment. The material meets the requirement for welding, machining, and the fabrication of mechanical tubing in a lower pH medium [24]. It can be the exemplary steel for designing corrosion inhibitor formulations (CIFs) since the characterizations of corrosion properties are similar to other low-grade steels for well-constructed materials.

Despite its excellent mechanical properties, the material is vulnerable to disintegration due to several factors. The work of [25] elaborates that the internal corrosion of pipelines leads to scale formation that may occur due to acidic HCl, which is pumped to improve the well's permeability and porosity. In addition, the research of [26] shows that the biofilm growth of Pseudomonas aeruginosa as a nitrate-reducing bacteria is accountable for a more corrosive C1018 without a sufficient organic carbon source.

Therefore, protecting the C1018 metal from the induced corrosion factor is essential. A recent study by [27] shows that applying pyridine green corrosion inhibitor is a practical and cost-efficient method of protection due to the presence of delocalized π-electrons and electro-negative functional groups that induce extensive adsorption on the C1018 metals. Furthermore, the study of [28] elaborates on the synergistic effect between 1-acetyl-3-thiosemicarbazide (AST) and iodide ions to control the corrosion rate of the same material under the acidic condition of 1 M HCl.

Some corrosion tests are significant in studying the development of a passive layer that increases corrosion resistance under more aggressive conditions. It includes electrochemical tests such as potentiodynamic polarization and electrochemical impedance spectroscopy (EIS). It is also important to note the use of machine learning allows the analysis of the efficiency of the inhibitor from the corrosion test to map the relationship between the input and output data. Our previous publication [29] shows the contribution of phenolic compounds in liquid smoke RHA at higher concentrations to increase the impedance and Nyquist diameter using EIS on the non-C1018 carbon steel. In the same paper, the exploration using the machine learning algorithm of Pearson Multicollinear Matrix successfully utilizes the electrochemical test datasets to model the inhibition mechanism.

The study of [30] argues that the neural network is suitable to model the corrosion since $CO_2$ as the confidence limit of test results stands at 95%. In the same study, the temperature, partial pressure of $CO_2$, pH, and corrosion rate serve as input data to predict the high partial pressures of $CO_2$ corrosion as output data. Nevertheless, the study to uptake the result of the corrosion test is somewhat limited because the different outcomes of the electrochemical tests at various concentrations and temperatures represent the different adsorption responses and passive film thickness. Using artificial neural networks (deep learning) is attractive in this work. They provide high accuracy to lower the variety of inhibition prediction results based on the electrochemical tests while keeping minimal human intervention and subjectivity. Notably, the trained dataset identifies the inhibition process pattern, while the test data estimates the numerical accuracy of the predicted passive film thickness. Thus, this paper intends to combine the experimental results, surface analysis, and the utilization of deep learning to acquire the corrosion test data and integrate it into a modeling system for a quicker adaptation to other corrosion inhibitor development scenarios.

## 3. Materials and Methods

### 3.1. Preparation of Working Electrode

The working electrode was made of mild steel C1018 with a thickness of 0.1 mm and was utilized for potentiodynamic polarization and electrochemical impedance study. The steel has a chemical composition (%wt) of 0.198 C, 0.710 Si, 0.9046 Mn, 0.151 Cr, and the balance Fe. The working electrode was sanded using Emery paper from 200 to 2000 grit to ensure cleanliness from rust and debris and increase the surface area of exposure. Before immersion in the sample solution (inhibitor and acidic solution), the steel was washed using distilled water and ethanol and dried in air.

### 3.2. Preparation of Inhibitor and Test Solution

The inhibitor solution was prepared using the pyrolysis process, which was published in our previous work [15]. In this study, the aggressive solution of 1 M HCl was prepared from 37 percent of the analytical-grade solution (Merck. Co, Darmstadt, Germany). The prepared acidic solution was used as a test solution for 20, 40, and 80 ppm concentrations of liquid smoke solution. Meanwhile, the HCl 1 M solution was taken without an inhibitor as a blank solution for comparison. The selected and prepared extract concentration was utilized for the electrochemical measurement. This work was also carried out at various temperatures of 303–323 K to evaluate the thermal stability of the inhibitors.

### 3.3. Electrochemical Method

The corrosion rate and the anti-corrosion behavior of mild steel at various concentrations and temperatures were measured using a Gamry G-750 Series, Gamry Instrument, Warminster, PA, USA. The working electrode of C1018 was mounted with a 1 cm × 1 cm exposed area, equipped with two-electrode cells. The auxiliary electrode was a platinum wire, and the reference electrode was a saturated calomel electrode (SCE). The working electrode was cleaned using various grades of emery paper and dried before starting the measurement. The three-electrode system was immersed in the test solution to record the

open circuit potential ($E_{OCP}$). The Tafel measurement was used to evaluate the corrosion rate at a range potential of $-250$ to $+250$ mV with a scanning rate of 1 mV/s according to ASTM G 59 standard [31]. The inhibition efficiency was calculated using Equation (1).

$$\eta = \frac{R_{inh} - R_s}{R_{inh}} \times 100\% \tag{1}$$

In the above equation, $\eta$ is the inhibition efficiency (%), $R_{inh}$ ($\Omega$) and $R_s$ ($\Omega$) are the resistance in the presence and without an inhibitor. The Bode and Nyquist plots were obtained using the electrochemical impedance measurements (EIS) of Gamry G-750 Series, Gamry Instrument, Warminster, PA, USA using a frequency of 100 kHz to 0.2 Hz with an AC (Alternating current) signal at 10 mV and the data were recorded using 10 points/decade. EC-Lab (version V10.40) and ZView software (version 3.4e) were used to carry out Tafel and EIS analysis in their corresponding frequency ranges.

### 3.4. UV-Vis and FTIR Analysis

The chemical structure of pure and film-formed LS extract characterization was confirmed using UV-Vis (Shimadzu/UV-1201, Shimadzu Corp., Kyoto, Japan), and FT-IR (Thermo Scientific Nicolet iS-10), Thermo Fisher Scientific, Waltham, MA, USA on the surface of mild steel, and the adsorption mechanism was discussed. The mild steel was immersed in a test solution comprised of 80 ppm LS extract for two hours before being removed from the solution and dried under nitrogen gas. Depending upon the structure, it was assumed that 2-methyl-phenol, 5-methyl- 2-furancarboxaldehyde, octadecamethyl-cyclononasiloxane, and 2-methyl-pyridine were accountable for playing a pivotal role in increasing the anti-corrosion behavior of C1018. The details of the results were published in [15], which verified the existence of the major compounds.

### 3.5. Surface Analysis

The cleaned working electrodes were immersed in 1 M HCl in the absence and presence of 80 ppm LS solution for 24 h at given temperatures. An Atomic Force Microscope (AFM) was utilized to probe the morphology of untreated and treated working electrodes on the surface. The 3D images and 2D height morphology to measure the roughness of the sample were obtained through a scanning technique using an NX10 Atomic Force Microscopy Park System, Park Systems Corp. Suwon, South Korea.

### 3.6. Deep Learning Studies

In this work, an artificial neural network (ANN) was used to evaluate the film thickness evolution on the metal surface at a higher temperature (323 K). The input data were generated from Nyquist, Bode Plot, and the Bode Phase at various concentrations. Before modeling, the preprocessing dataset was implemented to remove the irrelevant and null data. The unrelated data were removed to reduce computational complexity and increase the data processing accuracy. The implementation of feature-selection coding was used. Continuing to smooth and normalize data shows the actual thickness film evolution, which models the relationship between the increase in impedance, Bode phase angle, the potential of corrosion, corrosion current density, and the diameter of the adsorbed particles at a higher temperature. Before modeling, the dataset (see Supplementary Data S1) was split into training and testing data in portions of 80% and 20%. The output data were the diameter of the adsorbed particles, which appears as "0" for not developed and "1" as a developed passive film. All features (the input data) were passed to the hidden layer and activated using the activation functions of "ReLu" and "Softmax" in the output layers. The selected modeling architecture was set to be (8–25–2) and (24–5–2) for uninhibited and inhibited mild steels to achieve maximum iterations. The epoch number implementation of 100 s provides sufficient processing modeling time and converges the model. The work was completed using Python 3.7 stored in a personal computer, Intel iCore i7, RAM 8 GB.

### 3.7. Diameter Particle Measurements

The objective of obtaining the diameter of particles is to study the relationship between increased film thickness as the more concentrated solution was added and the diameter of adsorbed particles. In this research, the AFM image at 80 ppm 323 K was selected to represent the modeling and generate the output datasets. The open software Image-J (version 1.53 k) was used to analyze the area of the adsorbed particles and is assumed as a circle shape. The preprocessing technique includes the selection of the most representative film from the image and utilizing a bandpass filter to obtain the accurate conditions of inhibitor particles. The range of particles was five nm–infinity before the particle diameter range was obtained and used as output data for the classification of the ANN.

## 4. Results and Discussion

### 4.1. Anti-Corrosion Behavior of Liquid Smoke Inhibitor

The anti-corrosion performance of liquid smoke in the absence and presence of an inhibitor was analyzed using Tafel polarization and electrochemical studies. Figure 1a shows the trace of corrosion current density ($i_{corr}$) and their respective corrosion potential ($E_{corr}$), using Tafel to extrapolate the cathodic and anodic curves (βa and βc), which tabulate the results in Table 1. Equation (2) shows the formula for Tafel inhibition efficiency [32],

$$\eta(\%) = \left( \frac{i_{corr,free} - i_{corr,inhb}}{i_{corr,free}} \right) \times 100\% \tag{2}$$

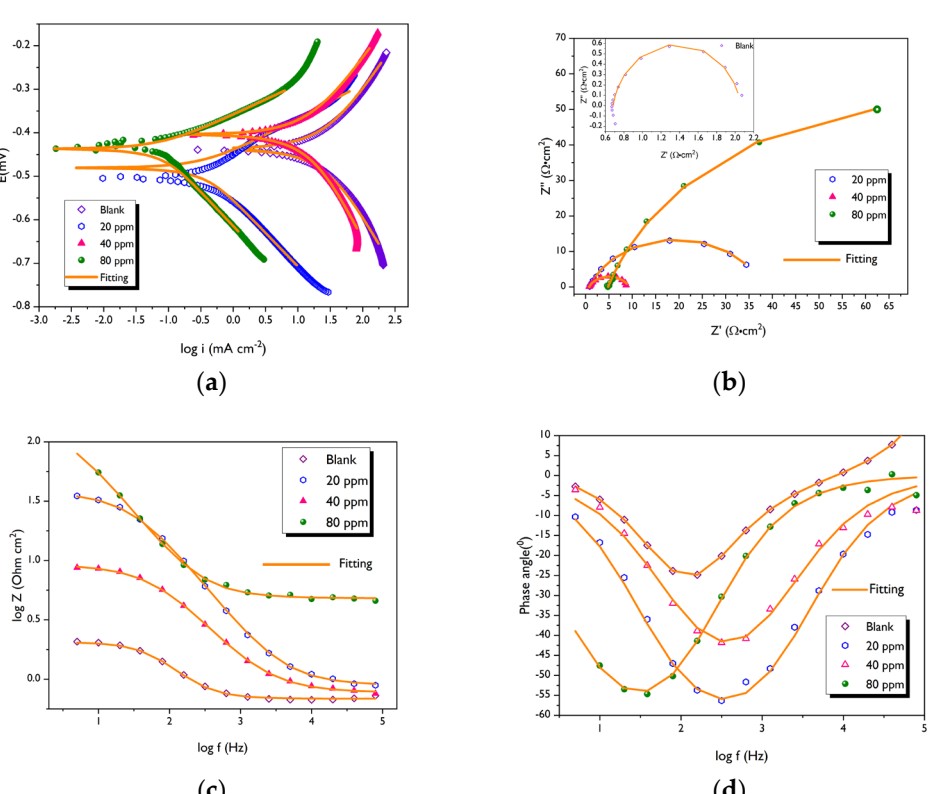

(**a**)

(**b**)

(**c**)

(**d**)

**Figure 1.** The anti-corrosion behavior of the inhibitor at 323 K. (**a**) Tafel polarization plots; (**b**) Nyquist plot; (**c**) Bode plot; (**d**) Bode phase.

**Table 1.** Results of potentiodynamic polarization.

| Conc (ppm) | Temp (K) | βa (mV/Decade) | βc (mV/Decade) | *Ecorr* (mV vs. SCE) | $i_{corr}$ (mA/cm$^2$) | $\eta$ (%) |
|---|---|---|---|---|---|---|
| Blank | 303.0 | 139.8 | 206.7 | −482.0 | 6.27 | − |
| Blank | 313.0 | 205.3 | 235.4 | −441.8 | 17.20 | − |
| Blank | 323.0 | 221.7 | 256.8 | −437.0 | 25.30 | − |
| 20 | 303.0 | 92.2 | 176.9 | −494.3 | 0.460 | 92.66 |
| 20 | 313.0 | 100.2 | 144.2 | −486.6 | 0.279 | 98.38 |
| 20 | 323.0 | 72.8 | 155.8 | −476.3 | 0.325 | 98.72 |
| 40 | 303.0 | 93.3 | 167.3 | −470.2 | 1.37 | 78.16 |
| 40 | 313.0 | 132.7 | 173.5 | −410.9 | 4.33 | 74.83 |
| 40 | 323.0 | 202.0 | 273.4 | −440.4 | 13.887 | 45.11 |
| 80 | 303.0 | 55.3 | 219.6 | −413.3 | 0.054 | 99.14 |
| 80 | 313.0 | 75.1 | 172.9 | −442.1 | 0.066 | 99.62 |
| 80 | 323.0 | 70.7 | 157.2 | −436.7 | 0.075 | 99.70 |

In the above equation, $i_{corr,free}$ and $i_{corr,inhb}$ are the corrosion current density in the uninhibited and inhibited solution.

The results of Figure 1a confirm the results of Table 1, which demonstrates the excellent inhibition of all inhibitor concentrations. Overall, adding an inhibitor solution results in the movement of $E_{corr}$ in both upward and downward directions, indicating that LS smoke inhibitor adsorption inhibits cathodic and anodic activity effectively (Table 1 and Figure 1a). At the cathode, the inhibitor works by the reduction of hydrogen ions. In contrast, the anodic region indicates that the inhibitor prevents metal dissolution (presumed pit formation). Based on the literature [33], the inhibitor is classified as a mixed type of inhibitor when the $E_{corr}$ for the inhibited system shift is less than ±0.085 V relative to the $E_{corr}$ value in the absence of an inhibitor. Since the value of $E_{corr}$ did not exceed the difference, the LS inhibitor is presumed to be a mixed type of inhibitor and confirmed by the gradual decrease in anodic current density [34]. Hence, this suggests the addition of an inhibitor forms a barrier layer that gradually grows on the C1018 surface. In addition, the shape of the Tafel plot is similar, suggesting that the inhibition mechanism has not been altered.

Few LS molecules are in the solution at a low concentration of 40 ppm, which fails to deliver the entire inhibition effect and gives higher corrosion current density (13.387 mA/cm$^{-2}$). It is also possible to presume that at the same solution, the direction of the inhibitor adsorbs in the vertical orientation, which makes it more difficult for corrosion inhibition to occur [35]. As a result, forming the passive film is more challenging to separate the metals and the solution to give lower inhibition efficiency at 45.11% and 323 K. On the other hand, adding more corrosion inhibitors into the solution results in low corrosion current density in both the anode and cathode curves and aligns with the trend in $E_{corr}$ [36]. The more concentrated inhibitor provides more significant participating functional groups in coating the bare metal, which results in lower electrochemical activity and raises the efficiency nearly two-fold. Based on the study of [37], a minor variation in βa and βc (Table 1) suggests that adsorption may occur through physical and chemical bonding. Hence, the above fact agrees well with the reported corrosion inhibition efficiency of 80 ppm and 323 K at 99.70%, which depresses the corrosion current density at 0.075 mmpy to reach the optimum protection and better shielding from the acidic solution.

### 4.2. Electrochemical Impedance Spectroscopy Results

It is noteworthy that the addition of corrosion inhibitors is inseparable from their inhibitory mechanism, where electrochemical impedance spectroscopy (EIS) was utilized. Figure 1b–d shows the EIS measurement (Nyquist, Bode plots, and phase angle plots) in the presence and absence of an inhibitor with various concentrations at 323 K. Figure 1b exhibits a half-loop of a one-time constant Nyquist plot corresponding to the charge transfer process at the surface of the working electrode [38]. Moreover, the diameter of single depressed

semicircles of the free-inhibitor solution is elongated for a non-free system to indicate a quicker charge transfer process.

Higher temperatures, including $Fe^{2+}$, promote metal dissolution in the system [39]. Therefore, it remains critical to note the presence of the LS inhibitor increasing the resistance of C1018 to corrosion, which is depicted by a more significant arc radius resistance of 80 ppm solution. It attributes to the formation of stable film protection to preserve the metallic surface and the surface heterogeneity under non-free inhibitor solutions. Likewise, the smaller arc of the blank solution exhibits a shorter semi-long axis to show the severe corrosion reaction occurrence. These results confirm the result of Tafel polarization, where the depression of corrosion current density is reduced in an HCl solution of 80 ppm at 323 K.

In addition, it is essential to note that the shapes of the curves are unmodified to imply similar electrochemical properties that coat the steel from corrodents. The similar pattern curve aligns with the polarization results (see Figure 1a) despite the imperfect semicircle plots in the 80 ppm solution. This may correspond to the dislocation and adsorption of the inhibitor or the surface roughness [40].

The Bode plots and the Bode phase agree well with the Nyquist plot to show that the charge transfer process occurs given the presence of one relaxation time, and the corrosion control is governed by charge transfer [41]. Figure 1c shows the Bode plot of the inhibitor at 323 K. In this context, the value of the absolute impedance of log |Z| for the inhibitor solution was nearly four orders of magnitude greater than the blank solution. The intercept at a lower frequency resembles the polarization resistance ($Rp$) [42]. Increasing concentration increases the absolute impedance while keeping a similar shape, corresponding to the inhibitory effect and greater capacitive response. On the other hand, Figure 1d shows the one-time Bode phase angle plot from the appearance of single phase peaks and broadens along the middle-frequency range. However, the extent of broadening is more pronounced in the inhibited system, where the phase angle is maximized to greater degrees. The changing trend of the phase angle into a negative value of less than 90° shows a better protective metal by smoothing the metallic surface and reducing the corrosion degree [43]. In addition, the more negative phase angle indicates more stable iron oxide through the adsorbed inhibitor on the metal surface and slower anodic reaction at all inhibited frequency ranges [44].

Furthermore, an equivalent electrical circuit (EEC) with a one-time constant fits the result of EIS that is commonly applicable for fitting the data on various metals [45,46] in studying the effectiveness of inhibitors (Figure 2).

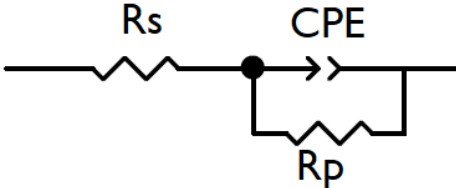

**Figure 2.** The equivalent electrical circuit for fitting and simulating the EIS data.

The circuit comprises $Rs$ as the solution resistance of the blank solution, CPE denotes the constant phase element, and $Rp$ is the polarization resistance of the entire film (passive film and inhibitor film). The passive film resistance is correlated with the charge transfer resistance, and the inhibitor film deals with the resistance of LS at various concentrations [47]. In the above circuit, the $Rs$ is in series with the CPE but simultaneously in parallel with the $Rp$. It is possible to use a one-time EEC constant that involves the contribution of the sole resistance of passive and inhibitor LS film inhibitor as the two-time constant model, which could provide insight and overlap in EIS responses, as depicted in Figure 1b–d [48,49]. Table 2 confirms the fitting of one-time EEC.

**Table 2.** The electrochemical parameters of mild steel immersed in the 1 M HCl and in the presence of rice husk ash inhibitor.

| Conc (ppm) | Temp (K) | CPE Component | | | $C_{dl}$ ($\mu F/cm^2$) | $R_s$ ($\Omega$) | $R_p$ ($\Omega$) | $\tau$ (mS) | $\eta$ (%) |
| | | $Y_0$ ($\times 10^{-3}$) ($\Omega^{-1}\, s^n\, cm^{-2}$) | $n$ | Chi-Sq | | | | | |
|---|---|---|---|---|---|---|---|---|---|
| Blank | 303.0 | $1.116 \pm 0.00018$ | $0.8875 \pm 0.0223$ | $16.8 \times 10^{-3}$ | 458.025 | $0.796 \pm 0.0158$ | $4.802 \pm 0.1760$ | 2.199 | - |
| Blank | 313.0 | $1.958 \pm 0.00047$ | $0.9026 \pm 0.0334$ | $6.2 \times 10^{-5}$ | 966.180 | $0.731 \pm 0.0159$ | $2.566 \pm 0.1227$ | 2.479 | - |
| Blank | 323.0 | $3.010 \pm 0.00101$ | $0.9063 \pm 0.0499$ | $5.3 \times 10^{-2}$ | 1588.451 | $0.685 \pm 0.0162$ | $1.373 \pm 0.0843$ | 2.181 | - |
| 20 | 303.0 | $0.660 \pm 0.00005$ | $0.7574 \pm 0.0099$ | $3.9 \times 10^{-3}$ | 67.008 | $1.195 \pm 0.0214$ | $26.09 \pm 0.7091$ | 1.748 | 81.59 |
| 20 | 313.0 | $0.335 \pm 0.00052$ | $0.7998 \pm 0.0099$ | $3.3 \times 10^{-3}$ | 44.753 | $0.960 \pm 0.0174$ | $62.05 \pm 1.7264$ | 2.777 | 95.86 |
| 20 | 323.0 | $0.388 \pm 0.00019$ | $0.7970 \pm 0.0006$ | $2.2 \times 10^{-3}$ | 51.052 | $0.888 \pm 0.0131$ | $36.96 \pm 0.7356$ | 1.887 | 96.29 |
| 40 | 303.0 | $0.305 \pm 0.00054$ | $0.8579 \pm 0.0248$ | $4.1 \times 10^{-2}$ | 80.190 | $1.144 \pm 0.0547$ | $56.26 \pm 4.7627$ | 4.512 | 91.46 |
| 40 | 313.0 | $0.686 \pm 0.00073$ | $0.8001 \pm 0.0139$ | $8.3 \times 10^{-3}$ | 106.492 | $0.844 \pm 0.0203$ | $19.13 \pm 0.70432$ | 2.037 | 86.59 |
| 40 | 323.0 | $1.057 \pm 0.00095$ | $0.7674 \pm 0.0115$ | $3.9 \times 10^{-3}$ | 122.482 | $0.771 \pm 0.0127$ | $8.523 \pm 0.19221$ | 1.044 | 83.89 |
| 80 | 303.0 | $3.676 \pm 0.00032$ | $0.7690 \pm 0.0143$ | $9.3 \times 10^{-3}$ | 1387.049 | $10.61 \pm 0.165$ | $233 \pm 19.067$ | 323.182 | 97.94 |
| 80 | 313.0 | $3.021 \pm 0.000526$ | $0.75195 \pm 0.0278$ | $34.4 \times 10^{-3}$ | 940.987 | $9.638 \pm 0.3488$ | $253.4 \pm 38.402$ | 238.446 | 98.99 |
| 80 | 323.0 | $0.553 \pm 0.00029$ | $0.8115 \pm 0.0089$ | $4.5 \times 10^{-3}$ | 139.537 | $4.812 \pm 0.0499$ | $137.8 \pm 7.3149$ | 19.228 | 99.00 |

In this work, the $C_{dl}$ (the ideal double-layer capacitance) displaces the CPE in the circuit, hindering the variation in frequency dispersion [50]. Equation (3) shows the formula of CPE [51],

$$Z_{CPE} = Q^{-1} \times (i \times \omega)^{-1} \tag{3}$$

where $Q$ is the constant imaginary *CPE* number, *i* is the imaginary number, and $\omega$ is the angular frequency of $2 \times \pi \times f$. Another critical piece of information from Table 2 is the value of $C_{dl}$, which is calculated using Equation (4) [52],

$$C_{dl} = (Y_0)^{(1/n)} \times (R_s)^{(1-n)/n} \tag{4}$$

In the above equation, $Y_0$ and *n* are the system passivity (surface modulus). The *n* value is a provision to explain the roughness and/or the heterogeneity factor of the mild steel surface. Table 2 shows that the importance of $C_{dl}$ in the presence of an inhibitor has considerably depressed the corrosion process at each observable concentration except for the 40 ppm solution. This indicates the reduced thickness of the double layer to allow more inhibitor molecules to displace water molecules at the interface of the mild steel/HCl 1 M solution [53]. Accordingly, the rise in *Rp* at 137.8 $\Omega$ and the ten-fold smaller value of $C_{dl}$ confirms an excellent inhibition efficiency at 99% and the reduction in surface roughness in the presence of liquid smoke inhibitors by increasing the electrical double layer thickness to coat the metal. In addition, the *n* value is closer to 1 (0.8115) and confirms the electrical circuit's capacitance constant phase element (CPE) (Figure 2) [54] and increases the surface homogeneity of the C1018 metals [55]. Additionally, the low value of the Chi-squared test shows that the fitted results agree well with the experimental outcomes.

Moreover, the result of the $C_{dl}$ calculation can be used to determine the relaxation time ($\tau$) of the inhibitor adsorption process, as shown in Equation (5) [56],

$$\tau = C_{dl} \times R_p \tag{5}$$

Table 2 shows that the relaxation time in the absence of an inhibitor and at lower concentrations is characterized by a shorter relaxation time. Likewise, increasing concentration is associated with a slower adsorption process as it lengthens the relaxation time [57].

### 4.3. UV-Vis and FTIR Results

4.3.1. UV-Vis Results

In this work, the pyrolysis condensation product of rice husk ash was prepared as an inhibitor, and the spectroscopic results are depicted in Figure 3.

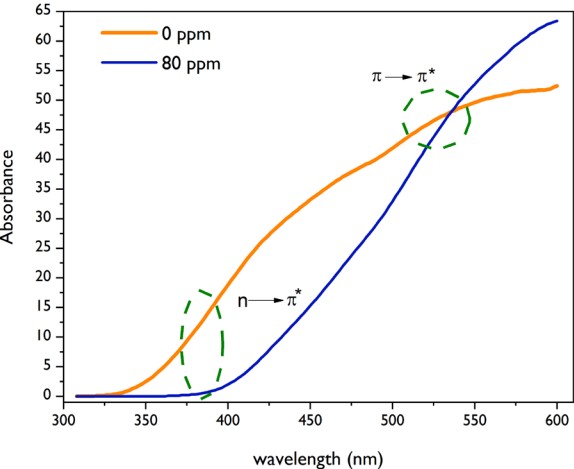

**Figure 3.** The UV-Vis spectroscopy result of the inhibitor solution.

The absorption spectra of liquid smoke pre- and post-immersion of mild steel in the 1 M HCl solution were recorded, and the results are depicted in Figure 3 within the 340–530 nm range. It is suggested that the intense peak in the UV-Vis spectrum is due to the chromophore's absorption in –OH (aromatic ring) compounds at approximately 530 nm and correlated to the conjugation of $\pi$–$\pi$* electronic transition. At the same time, the disappearance of a weak peak at 340 nm is associated with the transition electronics of n–$\pi$* transitions [58]. It is also noteworthy to observe that the peak of the blank solution weakens compared to the 80 ppm solution, which corresponds to the formation of chemical bonding due to the adsorption process between active molecules and $Fe^{2+}$ and $Fe^{3+}$ to form a more protective layer and complex molecules [59].

4.3.2. FTIR Results

The result of FTIR confirms the evaluation of the electronic transition of UV-Vis and is depicted in Figure 4.

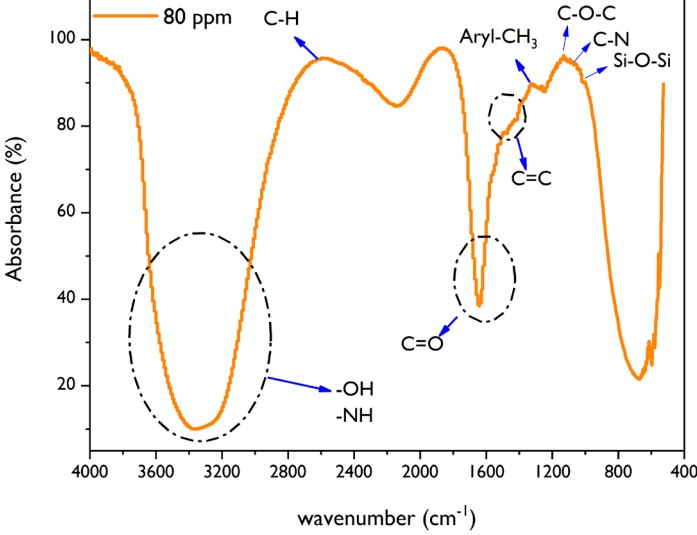

**Figure 4.** The FTIR results of liquid smoke inhibitor at 80 ppm.

The characteristic peak of FTIR shows an intensive and broad peak at 3000–3600 cm$^{-1}$, which might be due to –OH [60] and –NH bond stretching [61]. These functional groups allow the formation of hydrogen bonding between the substrate and adsorbate to increase inhibitor solubility in water. Accordingly, this results in a higher inhibitor resistance at 834.9 Ω and lowers the local dielectric constant. Both functional groups are predicted to bind with $Fe^{2+}$ and $Fe^{3+}$ by forming a dative covalent bond to give stronger chemical bonds [62]. The peak appearance at 2950 cm$^{-1}$ corresponds to the –CH stretching, consistent with that reported in [63]. The assigned peak at 1620 and 1495 cm$^{-1}$ is attributed to the C=O and C=C aromatic absorption [64].

In contrast, the molecules of Aryl–CH$_3$ and C–O–C correspond to the peak molecules at 1150 cm$^{-1}$. The ultimate contribution of the mentioned functional group is to increase the adsorption centers by donating pair of electrons on the 3D orbital of iron ions. Eventually, the published work of [65,66] shows that the absorption band at 1098 cm$^{-1}$ correlates to the asymmetric stretching vibration of Si–O–Si. Considering the appearance of the peak, the adsorption of Si–O–Si propagates the hydrophobic passive film of the 80 ppm solution to repel more water molecules [67]. The functional groups recorded from the UV-Vis and FTIR techniques are similarly linked to higher inhibitor performance (see Table 2).

### 4.4. Surface Analysis Studies

The Atomic Force Microscope (AFM) provides a valuable method to measure metal surface roughness after adding an inhibitor. This work immersed the mild steel in a blank solution and liquid smoke inhibitor at 80 ppm at 303 K for one day.

The AFM analysis is represented in 3D images and 2D height plots. It can be observed from Figure 5 that the surface of the mild steel was severely damaged through several corroded pit sites due to the deterioration of the working electrode. This is also evidence that the blank solution's average skewness value was 0.3281 nm. Nonetheless, after immersing mild steel in 80 ppm solution, the average value decreased to $-0.5190$ nm to indicate the protection of the inhibitor that was established against the corrosive HCl 1 M solution [68]. Furthermore, the maximum height peak roughness ($R_p$ and root mean square roughness ($R_q$) declined to 0.9907 nm and 0.5464 nm to indicate the surface treatment and improve the corrosion ability of the inhibitor.

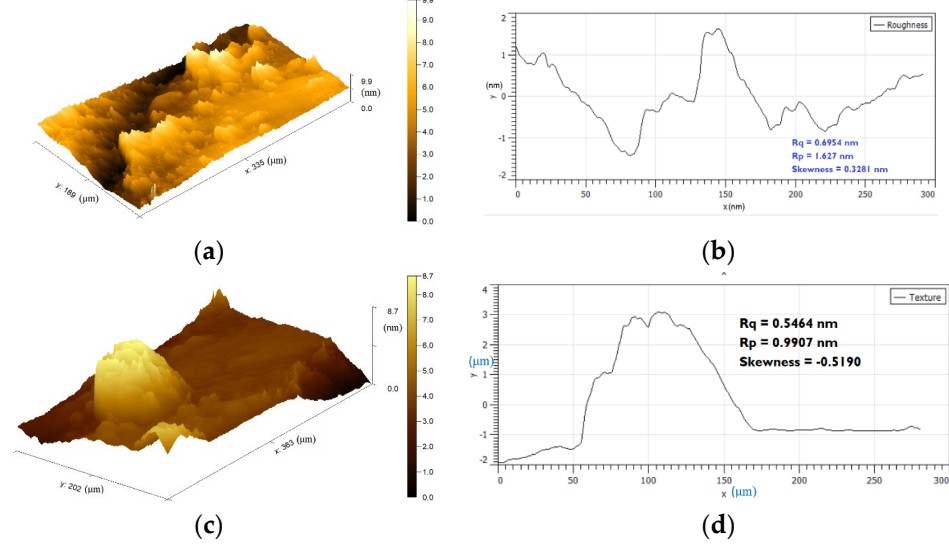

**Figure 5.** The AFM results of mild steel. (**a**) Uninhibited; (**b**) 2-D Height Plots of uninhibited; (**c**) Inhibited; (**d**) 2-D Height Plots of inhibited.

### 4.5. Deep Learning and Particle Size Studies

The neural network collects all the Tafel and EIS datasets to process the input data from EIS and Tafel measurements and to transfer the data for classifying the evolution of

the passive film. Table 3 presents the acquired data, such as the evolution of film prediction accuracy and loss, including the evaluated value to improve the agreement between the experimental result and the model predictions.

**Table 3.** The calculated accuracy of the passive film using ANN.

| Mode | Evolution Film Prediction Accuracy (%) | Evolution Film Prediction Loss (%) | Evolution Film Evaluation Accuracy (%) | Evolution Film Evaluation Loss (%) |
|---|---|---|---|---|
| Uninhibited | 66.67 | 4.6807 | 61.54 | 20.0593 |
| Inhibited | 81.08 | 0.6001 | 80.00 | 0.6032 |

All the data on the untreated steel show lower prediction and evaluation accuracy and the loss of the passive film thickness. The smaller value corresponds to the inadequate protection of bare metal from an acidic solution, which also appears in the lower value of $R_{inh}$ and higher corrosion current density, $i_{corr}$ (Tables 1 and 2) [69]. The trained and tested ANN predicts that the electrochemical activities decreased well when the applied inhibitor was directly adsorbed over the surface of metals, indicating a good adsorption process. Based on Table 3, the predicted and evaluated accuracy increased and lowered the loss rate, corresponding to a larger surface coverage area of protection to cover the active sites of mild steel by the liquid smoke inhibitors [70]. Moreover, the architecture of the ANN provided a partial solution, predicting only 953 instances correctly, although the evaluation of the model gives better performance. This fact also explains a more compact and uniform particle distribution in inhibited steel with a diameter of 100–200 nm (Figure 6d).

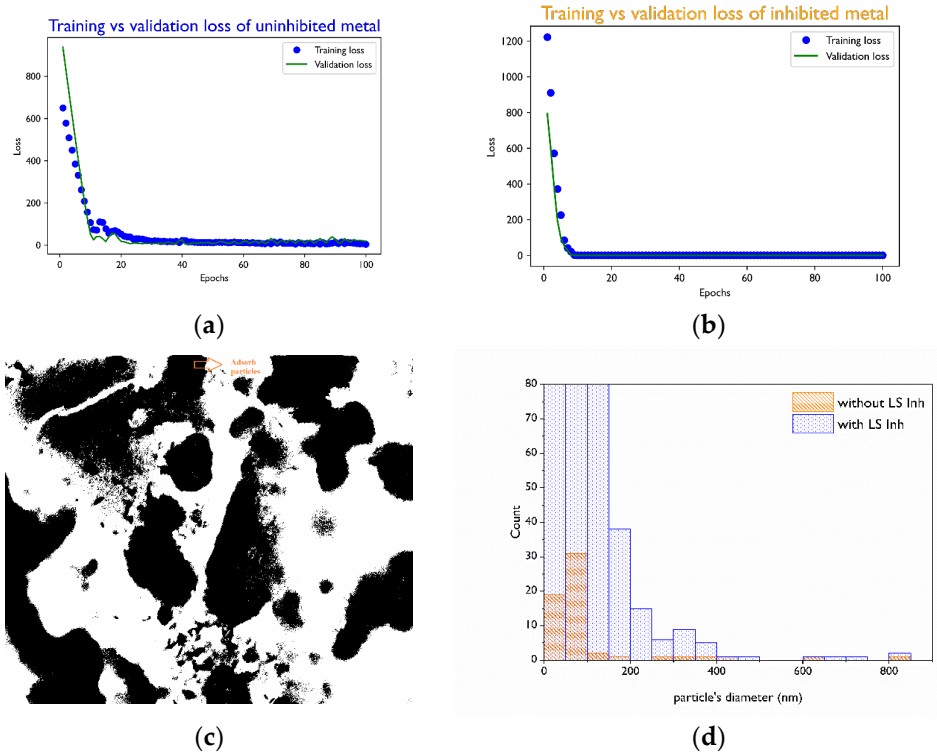

**Figure 6.** The ANN study and the diameters of adsorbed particles. (**a**) uninhibited metal; (**b**) inhibited metal; (**c**) distribution of adsorbed particles; (**d**) measured particle size before and after adsorption.

Table 3 shows the overfitting graph between the uninhibited and inhibited metals (Figure 6a,b). Considering Figure 6a, the overfitting of the training data agrees well with the lower accuracy of the model properties (66.67%), and it is unreliable to confirm the

formation of adsorbed film over the surface of metals. This model also aligns with the higher skewness parameter of untreated AFM measurements to give evidence of the mild steel surface roughness of 0.3281 nm and the lower number of the smaller particle diameter of the protective layer on the untreated metals.

On the contrary, the model for inhibited metal shows a better agreement between the training and validation loss. The overfitting in this model is low, and its result makes the forecasting about the evolution of film trustworthy. Based on Figure 6a, the non-linear relationship between the input data (EIS and Tafel) and the target (diameter of particles) data has a high correlation [71]. This result confirms the higher inhibition efficiency ($\eta$ = 99.7% and), as depicted in Figure 6c.

Therefore, the possible inhibition mechanism of an inhibitor (Inhb) can be proposed as follows:

$$Inhb_{(s)} \rightarrow Inhb_{(aq)}$$

$$Anode: Fe_{(s)} \rightarrow Fe^{2+}_{(aq)} + 2e^-$$

$$Inhb_{(aq)} + nFe^{2+}_{(aq)} \rightleftharpoons \left[ Inhb - \left( Fe^{2+}_{(aq)} \right)_n \right]$$

$$\left[ Inhb - \left( Fe^{2+}_{(aq)} \right)_n \right] \rightleftharpoons \left[ Inhb - Fe^{2+}_{(s)} \right]$$

$$Cathode: 2H^+_{(aq)} + 2e^- \rightleftharpoons H_2(aq)$$

The addition of the inhibitor reacting with hydrogen ions gives the adsorbed passive film.

$$Inhb_{(aq)} + H^+_{(aq)} \rightleftharpoons InhbH^+_{(aq)}$$

$$InhbH^+_{(aq)} \rightleftharpoons InhbH^+_{(s)}$$

Both reactions at the cathode and anode increase the adsorbed passive film.

The role of liquid smoke RHA inhibitors is evident in affecting the anodic region and influencing the liberation of hydrogen gas in the cathodic area (see Figure 1a). The addition of the inhibitor reacted with H$^+$ in the solution of HCl and became a film of inhibitor InhbH$^+_{(s)}$ at the cathodic site. The dissolved iron atom is bound with the inhibitor at the anode to give a dissolved complex $\left[ Inhb - Fe^{2+}_{(s)} \right]$. As a result, the inhibitor builds a thin film that protects the anodic region and slows the corrosion reaction by increasing the $E_{corr}$ to extend a more significant anodic shift (see Table 1). Moreover, the growth of the passive layer agrees with the result of the FTIR of Fe-Inhb in the adsorbed layer formed by liquid smoke RHA on the C1018 surface (see Figure 7).

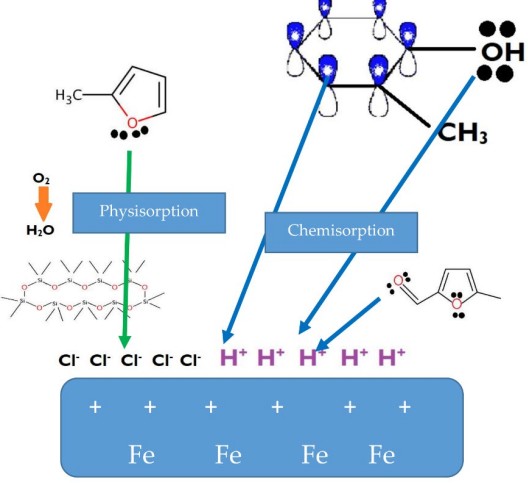

**Figure 7.** The schematic of the inhibition mechanism of inhibitors.

The successful collision between the adsorbed inhibitor and $Fe^{2+}/Fe^{3+}$ particles increases the number of the larger diameter of particles after the addition of the inhibitor, suggesting the growth of passive film, which is evident in treated mild steel (Figure 6d).

Based on the above evidence, the chemical and physical interactions can enhance RHA's anti-corrosion. The availability of unpaired p-electrons on the inhibitor's oxygen (hydroxyl) and nitrogen (amine) atoms increases the feasibility of forming a dative covalent bond (chemical bonding) between the d-orbital of iron and the inhibitor molecules through the chemisorption adsorption process. Moreover, the delocalization electrons on the benzene ring form a similar bond with a 3D vacant orbital of Fe over the surface of carbon steel [72].

Accordingly, it is possible to presume that the physical interaction occurred between the anion of chloride ions and protonated Lewis base of nitrogen. Similar to Berdimurodov [73], the excess hydrogen ions of HCl are diminished by the reaction with the hydroxyl functional groups from the active compound of RHA. The result of the Tafel plot related to anodic and cathodic branches confirms the pair of bonds (physically and chemically) (see Table 1). Therefore, it may be concluded that the LS smoke of RHA mitigates the corrosion of C1018 by adsorbing homogeneously and forming a passive film by repelling more water molecules.

## 5. Conclusions

In this work, the attempt to unveil the anti-corrosion properties of selected molecules provides a new insight to predict the inhibition reaction. The electrochemical and potentiodynamic results demonstrate that the liquid smoke inhibitor reduces the corrosion rate to show a maximum inhibition at 80 ppm, which is suitable for field applications. As a result of inhibition, the corrosion current density is lowered, governed by the charge transfer process. Increasing inhibitor resistance and efficiency reduces the double-layer capacitance in a more concentrated inhibitor solution. Moreover, it indicates the formation of a dative covalent bond between the metals and inhibitor. The UV-Vis spectra elucidated from the electronic transition of π–π* and n–π* can be attributed to the absorption of the liquid smoke inhibitor. FTIR agrees with the UV-Vis result to provide clear evidence on the adsorption of functional groups such as –OH, –NH, C=O, Si–O–Si, and the delocalization of π-electrons in the benzene ring on the surface of C1018 carbon steel. The study on distribution particles and AFM reveal a considerable reduction in the surface roughness, and the passive film is well distributed on the damaged mild steel surface. The modeling results of the ANN show a promising method to assert the experimental results. The lower overfitting plot and higher accuracy between the prediction and evaluation model for an inhibited surface show the agreement between the training and testing data, illustrating the formation of a more compact passive film.

**Supplementary Materials:** The supplementary material can be downloaded at https://www.mdpi.com/article/10.3390/coatings13010136/s1.

**Author Contributions:** Conceptualization, A.P.S.K., J.W.S., A.M. and R.R.; methodology, A.P.S.K.; software, A.P.S.K.; validation, A.P.S.K., J.W.S. and R.R.; formal analysis, A.P.S.K. and M.S.A.; investigation, A.P.S.K., A.R., M.S.A. and W.M.; resources, A.M., A.R., M.S.A. and W.M.; data curation, A.P.S.K., A.R., M.S.A., W.M.; writing—A.P.S.K. and M.S.A.; writing—review and editing, A.P.S.K., M.S.A., R.R. and J.W.S.; visualization, A.P.S.K.; supervision, M.S.A. and R.R.; project administration, A.M. and A.P.S.K.; funding acquisition, A.M., J.W.S. and R.R. All authors have read and agreed to the published version of the manuscript.

**Funding:** Matching fund program of Kedaireka in the Year 2022 with contract number PKS-595/UN2.INV/HKP.05/2022.

**Institutional Review Board Statement:** Ethical review and approval were waived for this study for not involving humans nor animals.

**Informed Consent Statement:** Not applicable.

**Data Availability Statement:** The data presented in this study are available on reasonable request.

**Acknowledgments:** The author gratefully thanks the Ministry of Research and Technology/National Research and Innovation Agency of Indonesia.

**Conflicts of Interest:** The authors declare that they have no conflict of interest with this research, both financial and personal authorship, that could affect the quality of the published manuscript.

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
