# Peer review of "Insight on Corrosion Prevention of C1018 in 1.0 M Hydrochloric Acid Using Liquid Smoke of Rice Husk Ash: Electrochemical, Surface Analysis, and Deep Learning Studies"

_coatings, doi:10.3390/coatings13010136_

Round 1

Reviewer 1 Report

General comments
Journal: Coatings

Manuscript ID: coatings-2009203

Title: Insight on Corrosion prevention of AISI C1018 in 1.0 M Hydrochloric

acid using liquid smoke of rice husk ash: electrochemical, surface analysis,

and deep learning studies

    Specific Comments:

Title

1. The introduction is too long and lack of the illustration of the motivation and novelty of the study.

2. Why do authors choose Liquid Smoke of Rice Husk Ash? What is the anti-corrosion mechanism?

3. What is AISI C1018? There is lack of sufficient description of the necessary of protecting AISI C1018.

4. Which part shows the various of inhibitor concentration in Fig. 2?

5. Do the authors carry out experiment to verify the inhabitation mechanism? What are the main conclusions of this study?  

6. What is the role of Rice Husk Ash in enhancing anticorrosion performances?

Reviewer 2 Report

I have carefully examined the manuscript coatings-2009203 and I present you my questions below.

1)    In line number 177, the scanning rate of 1 mV/s is too high for corrosion studies.

2)    In figure 1a, the solution without liquid smoke inhibitor has a smaller anodic and cathodic current density. So, the inhibitor presence indicates an increasing the corrosion process. On the other hand, the IZI at lower frequencies was smaller for the steel in the solution without an inhibitor. I ask the authors to explain the contradictory results.  

3)    In figure 1b, I ask the authors to show the EIS results for a solution without an inhibitor.

4)    In line number 253, the sentence “The inhibitor covers the surface of the metal through the reduction process of Fe3+ to Fe2+ at all concentrations.” I do not understand this sentence. The reduction reaction in the acid solution is of the hydrogen ion.

5)    In line number 311, please I ask you to cite the reference for this equation.

6)    In table 2 and in the manuscript, it is necessary to present the parameters that indicate the fitting quality.

7)    In line number 403, it is not possible to determine ΔHads, ΔSads, and ΔGads for a system where the active compound is not known.

8)    The inhibitor composition has more than two compounds, so, it is not possible to determine the adsorption isotherm for the compound mixture.

9)    In line number 432, I did not find figure 5 in the manuscript.

Author Response

Please see the attachment. We have revised the manuscript and added the response to reviewer #2 at the end of the text.

Round 2

Reviewer 1 Report

The authors have addressed all of the queries.

Author Response

Thank you so much for your professional willingness in advancing our paper. we greatly appreciated. 

Reviewer 2 Report

I examined the new version of the manuscript coatings-2009203 and I present you my point of view on the article:

1) What is the fitting parameter used? If it was the χ2 value, this value must be on the order of 10^3 and the error values must be smaller than 30% for each element of the circuit (Please see the reference J. Ross Macdonald (1987) Impedance Spectroscopy. New York, New York).

2) The authors must give out more details about the equivalent electrical circuit fitting. For example, what is the software used in fitting?

3) The impedance units in figure 1 are not correct.

I think that the EIS result analyses are inadequate. So, I must reject the publishing of this article.

Author Response

Please see the attachment. The response to reviewer is attached and additionally will be forwarded by Ms. Rini Riastuti.
